# Sub-national modelling of surveillance sensitivity to inform declaration of disease elimination: A retrospective validation against the elimination of wild poliovirus in Nigeria

**Emily S. Nightingale**[1,2,*], **Ly Pham-Minh**[3], **Isah Mohammed Bello**[3], **Samuel Okrior**[4], **Tesfaye Bedada Erbeto**[5], **Marycelin Baba**[6], **Adekunle Adeneji**[7], **Megan Auzenbergs**[1,2], **William John Edmunds**[1,2], **Kathleen M. O'Reilly**[1,2]

**1** Department of Infectious Disease Epidemiology and Dynamics, Faculty of Epidemiology and Population Health, London School of Hygiene and Tropical Medicine, London, United Kingdom, **2** Centre for Mathematical Modelling of Infectious Diseases, London School of Hygiene and Tropical Medicine, London, United Kingdom, **3** Polio Eradication Department, World Health Organization, Geneva, Switzerland, **4** The Bill and Melinda Gates Foundation, Seattle, Washington, United States of America, **5** World Health Organization, Country Office, Abuja, Nigeria, **6** National Polio Laboratory, Maiduguri, Nigeria, **7** National Polio Laboratory, Ibadan, Nigeria

* Emily.Nightingale@lshtm.ac.uk

## Abstract

A fundamental question in the global commitment to polio eradication is how long a period of absence would be consistent with regional elimination, and the safe withdrawal of the oral polio vaccine is contingent on the answer. We present a statistical framework to model the time-varying sensitivity of two key components of polio surveillance - environmental sampling and clinical cases of acute flaccid paralysis - for detecting infection on a monthly basis at the local government authority level. We use this to estimate the probability of freedom from infection (FFI) at a critical prevalence level that is consistent with interruption of transmission, given the absence of virus in collected samples. We validated this framework against two periods of poliovirus absence in Nigeria (2014–2016 and 2016–2020). Our model highlights substantial heterogeneity in surveillance sensitivity over time and space and, given this, concluded an 85% probability (95% uncertainty interval: 77.1-90.0%) of the country being free from WPV1 infection after 23 months without detection from July 2014. Detection of WPV1 in July 2016 demonstrated that circulation had indeed persisted during this time. In contrast, we conclude a probability of 98% (97.5-98.5%) by the time elimination of the serotype was officially declared in 2020. The inferred probability of FFI during both time periods was found to be consistent with the retrospectively known status of regional elimination. This supports the validity of applying this framework prospectively to inform the certification of wild poliovirus elimination from remaining endemic regions, and to determine the resolution of cVDPV2 outbreaks.

which permits unrestricted use, distribution, and reproduction in any medium, provided the original author and source are credited.

**Data availability statement:** Detailed disease surveillance data on which this research is based are available from the WHO Institutional Data Access / Ethics Committee, for Global Polio Eradication Initiative research partners who meet the criteria for access to confidential data. Any requests should be directed to the GPEI Polio Research Committee at poliore-search@who.int. The code used to conduct this analysis is available in the following repository, alongside a test dataset of randomised values: https://github.com/esnightingale/polio-ffi-model.

**Funding:** This research is supported by grants from the Polio Research Committee (https://polioeradication.org; ref:1432457-1) (ESN, WJE) and the Bill and Melinda Gates Foundation (https://gatesfoundation.org; ref: INV-049298) (KMO, MA, WJE). The funders had no role in study design, data collection and analysis, decision to publish, or preparation of the manuscript. The remaining authors (LPM, IMB, SO, TBE, MB, AA) received no specific funding for this work.

**Competing interests:** The authors have declared that no competing interests exist.

## Author summary

This study addresses a critical question in global polio eradication: how long must poliovirus be absent before a region can be considered free from infection and the oral polio vaccine can be safely withdrawn? We developed a statistical model to assess the effectiveness of two main surveillance methods—environmental sampling and monitoring of clinical cases—for detecting poliovirus at a local level each month. Applying this model to data from Nigeria, we estimated how confident we could be that the virus was eliminated, given two periods without detection between 2014 and 2020. Our findings revealed that, by mid-2016 after 23 months with no poliovirus found, there was an 85% chance infection had fallen to elimination levels—but subsequent detection of further cases showed that actual virus circulation had persisted. By the time of the official declaration of elimination in 2020, this probability was 98%. Comparing these figures to what was later known about the presence of polio confirmed our approach's accuracy. This framework can help policymakers decide when a region can be considered polio-free and guide steps to end vaccine-derived outbreaks, moving us closer to a polio-free world.

## Introduction

Since the World Health Assembly's resolution for a global commitment to polio eradication, an important question has been how long of a period of absence would be consistent with local elimination. Confidence in the attainment of elimination must be inferred from surveillance data and our understanding of the system's sensitivity to detect circulating viruses. The latter is challenging to quantify for complex systems consisting of multiple components and data streams, such as that for poliovirus surveillance.

In contrast to smallpox (which was eradicated in 1980) poliovirus is known to cause asymptomatic infection, meaning that infection can effectively spread 'silently' within a population for some time until a paralytic case is reported. In a paper by Eichner and Dietz from 1996 [1], a transmission model of both symptomatic (causing acute flaccid paralysis, or AFP) and asymptomatic infections was used to estimate the probability of unobserved, asymptomatic infections with increasing time since the last observed paralytic case. The authors concluded that a three-year time period would be sufficient to reach a 95% probability that the number of infections had been brought to zero. This inference was utilised by the Global Polio Eradication Initiative's Global Certification Commission (GPEI, GCC) to define initial criteria for declaration of national polio elimination. The modelling, however, assumed that all paralytic cases would be reported and did not account for environmental surveillance (ES) for circulating virus in wastewater, which is now an important tool in at least 45 countries [2].

More recent analyses have accounted for a more nuanced combination of surveillance strategies. Assuming varying degrees of "quality" of both AFP and ES [3–7] is

found to strongly influence confidence in interruption of transmission given an absence of observed positives. The GCC's interpretation of elimination therefore takes into account sub-national indicators of surveillance quality, which are monitored with respect to defined thresholds for "adequate" performance. Key indicators of interest are the rate of reported non-polio AFP among children under 15 (aiming for at least 2 per 100,000 in endemic countries), the proportion of reported AFP cases for whom adequate stool sample is collected (at least 85%), and the detection rate of enteroviruses in collected wastewater samples (at least 50%) [8,9]. The fact that these indicators do not directly translate to a formal definition of sensitivity, and vary substantially over time and space, complicates interpretation of what performance is adequate to conclude elimination.

Several previous studies describe the "sensitivity" of polio surveillance, but few refer to the formal definition of the term. Sensitivity is referred to as a status achieved by meeting the WHO-defined thresholds for the above indicators [10,11], or by the presence of correlation between vaccine doses distributed and detection rate in collected samples [12]. The detection rate of enteroviruses in ES has been used as a relative measure of sensitivity with which to rank sampling sites [5], and the relative sensitivity of ES and AFP explored according to which method detected an outbreak or cluster first (or at all) [13].

Ranta [14] and O'Reilly [4] defined epidemiological models of polio transmission to infer a conditional probability of detection given infection status - the former via a simulated population and the latter fit to observed data from Pakistan - and Kroiss [6] defines a similar probability via the detection rate of vaccine virus at ES sites in proximity to known vaccination activities. Watkins [15] employed a scenario tree approach to model the AFP surveillance system and estimate its sensitivity relative to an assumed infection prevalence. Although the latter four studies all aim for the formal definition of sensitivity as opposed to a proxy indicator of surveillance performance, differences in definitions, assumptions, context and data sources result in highly variable estimates, from 1-30% for AFP surveillance and 30–55% for ES.

If surveillance can be quantified with respect to the probabilistic definition of sensitivity for detecting infection, this can be inverted via standard probability rules to obtain the probability of a negative outcome given absence of infection. The trajectory with which this conditional probability increases with each month of negative observations can provide a timeline for declaring elimination. This approach, termed 'freedom from infection' (FFI), was proposed by Martin et al. in 2007 [16] has been applied widely in veterinary medicine [17–21] and for a number of human infectious diseases, including polio [4,15,22–25]. Here, we apply and validate the estimation of regional freedom from wild poliovirus infection using the retrospective example of wild poliovirus serotype 1 (WPV1) elimination in Nigeria.

Prior to 2018, Nigeria was endemic for wild poliovirus. Elimination was almost declared in 2016, when WPV1 had not been detected in any local government authority (LGA, equivalent to district) for a period of nearly two years [26,27]. A modelling study based on data up to 31st March 2015 concluded an 84% probability that WPV1 had been eliminated in Nigeria, and that if elimination *hadn't* been achieved a case would be detected with >99% probability by the end of 2015 [28]. This study crudely accounted for imperfect surveillance sensitivity by reducing the assumed case-to-infection (of 1:200) ratios by 50%, and did not account for evidence gathered from ES. In fact, WPV1 re-emerged in July 2016 in four clinical cases and four healthy contacts, all within the northeastern state of Borno. No evidence of WPV1 has been detected through either AFP or environmental surveillance since 2016, and interruption of transmission was officially certified in August 2020 [27].

We describe and apply a scenario tree approach to model the sensitivity of surveillance and estimate the probability of FFI for Nigeria during these two significant periods without detection of WPV1 (2014–16 and 2016–20). We consider the evident persistence of circulation as of July 2016 and the official elimination as of August 2020 as known outcomes, against which we validate our inferred confidence in FFI. We illustrate the monthly accumulation of evidence from negative surveillance observations, conditional on time-varying surveillance sensitivity during these two periods, and discuss what value this approach offers for the GCC's declaration of global poliovirus eradication.

## Materials and methods

### Ethics statement

This study was approved by the ethics committee of the London School of Hygiene and Tropical Medicine (approval number: 29381, granted 18th April 2023). Individual consent was not necessary to obtain due to anonymity of routinely-collected case notifications.

### Polio surveillance and data collection

Children under the age of 15 years who present to primary care with sudden weakness or paralysis (i.e., AFP) in their limbs should be referred for stool testing to determine whether poliovirus infection is present. Every notification of AFP is recorded by the national polio surveillance programme and entered into the GPEI's Polio Information System (POLIS) [29]. It is flagged as to whether the patient had adequate stool collected for analysis - defined as two separate samples collected > 24h apart, within 14 days of onset of paralysis and which arrive at the lab in good condition [30] - and this should be achieved for at least 85% of cases. The outcome of the stool analysis defines the case as either non-polio AFP - the vast majority, or a clinical case of paralytic polio.

Samples of wastewater are collected regularly from defined sites ranging from formal sewage treatment plants to open drains and natural waterways. Sites were initially selected with the intention to capture a population of approximately 100,000. In practice there is no guarantee that a site reflects this intended catchment size, which is likely to vary substantially depending on the positioning of the site and population movement since its initiation. We therefore estimate a catchment size for each site based on recent estimated population count within a fixed radius, instead of assuming that sites meet this target. This approach was shown by Hamisu et al. [5] to correlate more strongly with enterovirus prevalence (an indicator of catchment size) than estimates reported by ES officers.

Collected samples are concentrated and analysed for the presence of both polio and non-polio enteroviruses (EV). Samples are refrigerated for up to 3 days or frozen if more time is needed, and arrive at the lab within 3 days of collection, with adequate quality interpreted as an overall EV detection rate of at least 50% [2]. The location, timing and outcomes from each sample are also recorded within the POLIS system.

For the present analysis, reported cases of AFP and collected environmental samples between January 2013 and August 2020 in Nigeria were obtained from POLIS. We aggregated these data to a monthly time scale and a spatial scale of the local government authority (LGA; administrative level two), of which Nigeria has 774. Gridded (100m²) population count estimates were obtained from WorldPop [31], for inference of proportional population catchment around environmental surveillance sites in each LGA. The proportion of under-15s per LGA was taken from sex and age-specific estimates for the time period of interest, projected from the 2006 census [32].

### Immunity modelling

Monthly estimates of mucosal immunity against paralytic poliomyelitis per LGA were estimated using the same methods as used in a previous analysis of cVDPV2 outbreak risk on a sub-national level [33]. These are informed by a regression analysis of caregiver-reported vaccine doses relative to recent Supplementary Immunisation Activities (SIAs) among reported non-polio AFP cases across 47 countries, and estimates of routine immunisation coverage from the Institute for Health Metrics and Evaluation (IHME) [34].

### Statistical framework

Our goal is to understand whether poliovirus is truly absent from a region when no evidence of infection has been observed through the surveillance system. Letting $T^-$ denote the *observed* absence of infection and $I^-$ the underlying *true*

absence of infection, then the value of interest is the conditional probability $P[I^-|T^-]$. This is known as the negative predictive value (NPV), in the medical literature often interpreted for an individual diagnostic test $T$:

$$NPV = P\left[I^-|T^-\right] = \frac{True\ negatives}{True\ negatives + False\ negatives}$$

Using Bayes' theorem to invert the conditional probabilities, the NPV is linked to the sensitivity $Se$ and specificity $Sp$ of the test and to the underlying prevalence ($p$) of the condition being tested for:

$$NPV = \frac{Sp \cdot (1-p)}{Sp \cdot (1-p) + (1-Se) \cdot p}$$

Where

$$Se = P\left[T^+|I^+\right] = \frac{True\ positives}{True\ positives + False\ negatives}$$

and

$$Sp = P\left[T^-|I^-\right] = \frac{True\ negatives}{True\ negatives + False\ negatives}$$

following the standard definitions of sensitivity and specificity.

Here, we follow the approach from [16], extending this concept to a population level and construct a multi-stage, multi-component model of a "test" that represents the complexity of the polio surveillance system. In this scenario, a negative result is defined when *all* AFP cases and environmental samples tested within a particular time step are found to be negative for poliovirus. Conversely, a positive result is defined if poliovirus is detected in *any* tested sample.

### Freedom from infection

By assuming a critical prevalence level which would be consistent with interruption of transmission (the *design prevalence*, $Dp$), quantifying the probability of a positive result from this test given such a prevalence (the test sensitivity, $Se$) and assuming that false positive results are not possible (all positive results are followed up to complete confirmation, resulting in test specificity, $Sp$, of 100%), we are able to calculate the NPV defined above. This is interpreted as the probability that a region is genuinely free from infection above the design prevalence (i.e., elimination has been achieved) during a particular time step, given that the surveillance system has returned a negative result. This probability is defined as the *probability of freedom from infection*, or $P^{FFI}$.

We specify the design prevalence at two levels: LGA and country. There is not concrete evidence for a magnitude of infection prevalence that would sustain transmission, therefore we chose a value of $Dp = 1/100,00$ as well as two less-stringent thresholds to explore in a sensitivity analysis. This was a pragmatic choice on the basis that the threshold should be at least less than 1/2,000 (due to the asymptomatic rate of WPV1) and conservative (due to the significant negative consequences of premature declaration). We then set a country-level, or *regional*, design prevalence $Dp^{region} = 1/774$ such that if prevalence persisted above $Dp = 1/100,00$ in any <u>one</u> of the 774 LGAs, we would not interpret the country as a whole to be free from infection.

### Modelling surveillance sensitivity

As presented above, the general definition of sensitivity is the conditional probability that a test ($T$) yields a positive result, given that the condition being tested for ($I$) is truly present. To estimate this probability for the entire surveillance system as

a "test" for underlying poliovirus infection, we break down the system's two components (AFP and ENV) into the sequence of events that lead to a positive result, i.e., a reported detection of poliovirus (Fig 1). We assume that these surveillance components operate independently, and are therefore treated as independent probabilities.

The assumed stages of detection of poliovirus via AFP surveillance are as follows: an infection must first develop into clinical disease (i.e., with symptoms of paralysis; $P^{Clilincal}$), then such symptoms must be recognised and presented to a healthcare provider ($P^{Notified}$). The provider must collect adequate stool samples from the patient ($P^{StoolAdeq}$) to be sent for testing. Finally, poliovirus must be detected in the sample via lab analysis ($P^{StoolTest}$).

For environmental surveillance, an infected individual will only be captured if they are defecating and shedding virus into the wastewater within the catchment area of an active site ($P^{Catchment}$). The duration of shedding must coincide with collection of a sample at that site ($P^{Sampled}$), and the sample collected must be of adequate quality to detect enteroviruses ($P^{SampleAdeq}$) Finally, as before, presence of the virus must be detected via lab analysis (assumed to be virus culture and intratypic differentiation as per the protocol for the Global Polio Laboratory Networks [35]; ($P^{SampleTest}$). We define the catchment of ES as the area within 5km of an actively sampling site, within the boundary of the LGA in which the site falls. This definition excludes areas within 5km which extend into neighbouring LGAs, to which no samples are attributed. This will result in some under-estimation of the overall coverage of ES, but allows estimation of sensitivity within the specific LGA in which ES is active. A supplementary analysis explores the influence of the choice of catchment radius.

Given this set of events, we define the individual-level sensitivity as the probability that a *single* infected individual would be detected through AFP surveillance at time $t$ in LGA $i$ as

$$P_{it}^{unit,AFP} = P^{Clinical} \cdot P_{it}^{Notified} \cdot P_{it}^{StoolAdeq} \cdot P_{it}^{StoolTest}$$

and through ENV surveillance,

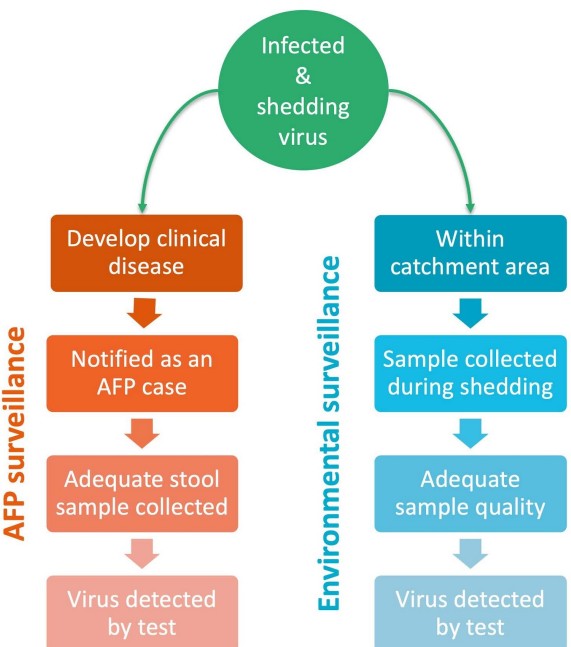

**Fig 1. Illustration of the two components of the polio surveillance system - AFP surveillance and environmental surveillance - and the sequence of events which must occur for a single infection to result in a reported detection of poliovirus.**

$$P_{it}^{unit,ENV} = P_i^{Catchment} \cdot P_{it}^{Sampled} \cdot P_{it}^{SampleAdeq} \cdot P^{SampleTest}$$

using the same notation. A distribution for the probability of each of these events is specified, either based on assumptions from the literature or estimated from recent data in the LGA. Each parameter is attributed with some uncertainty, either defined by a previously published uncertainty interval or based on a standard binomial distribution where the parameter is calculated from prior data. This uncertainty is propagated through subsequent calculations to obtain similar uncertainty intervals for sensitivity and FFI probability estimates. A detailed explanation of how the probabilities for each event were specified is given in the *Supplementary Materials*, and a summary is provided in Table 1.

### Sensitivity per LGA

The LGA-specific sensitivity of AFP and ENV surveillance is the probability that any infected individual in the assessed population will yield a positive outcome, given that there are infected individuals in the population at the assumed design prevalence, $Dp$. Here, we consider the number of individuals "assessed" by the surveillance system, $N_i$, to be the under-15 population size. Although the ES system in theory captures all ages, those older than 15 years have minimal risk of poliovirus infection due to built-up immunity [38]. They would have minimal influence on sensitivity since their probability of being infected is negligible, therefore we chose to exclude the older population from our calculations.

Assuming all under-15s in the LGA as independent and that risk of infection is homogenous within each LGA, the probability of *any* assessed individual yielding a positive outcome (either through AFP or ENV), given that each is infected with probability $Dp$ is

$$Se_{it}^{AFP/ENV} = 1 - (1 - Dp \cdot P_{it}^{unit, AFP/ENV})^{N_i}$$

since we assume a specificity of 100%.

**Table 1. Specification of event probabilities within each surveillance component. Each event must occur for a given infected individual to result in a reported detection of poliovirus. For all parameters except catchment of ES, a Beta distribution is fit to the given value and uncertainty interval from which random draws are used in the subsequent analysis. Uncertainty in the catchment population is explored through a sensitivity analysis of the assumed radius.**

| Surveillance component | Event | Fixed/ Time-updated | National/ LGA-specific | Estimate (95% uncertainty interval, where applicable) |
|---|---|---|---|---|
| **AFP** | An infection develops into clinical case of AFP: $P^{Clinical}$ | Fixed | National | 0.005 (0.004 - 0.006) [36] |
| | A clinical case of AFP is notified to the health system: $P_{it}^{Notified}$ | Time-updated | LGA-specific | 0.9 (0.6 - 0.99) *Scaled by LGA rate of reported AFP relative to 2/100,000 target, prior 12m.* |
| | Adequate stool sample is collected from the case for laboratory testing: $P_{it}^{StoolAdeq}$ | Time-updated | LGA-specific | *Proportion of reported AFP cases with adequate stool sampling, prior 12m (with 95% binomial confidence interval).* |
| | Laboratory testing detects poliovirus in the sample: $P^{StoolTest}$ | Fixed | National | 0.97 (0.95 - 1.00) [37] |
| **Environmental** | Infected individual is within the catchment area of active ES site(s): $P_{it}^{Catchment}$ | Time-updated | LGA-specific | *Proportion of LGA population within 5km of currently active ES site(s), with uncertainty interval of +/- 15%* |
| | A sample is collected while the individual is shedding virus: $P_{it}^{Sampled}$ | Time-updated | LGA-specific | *Median and 95% quantile interval across simulations, given mean sampling frequency, prior 12m.* |
| | Collected sample is of adequate quality to detect virus: $P_{it}^{SampleAdeq}$ | Time-updated | LGA-specific | *Proportion of samples in which EV was detected per LGA, in the last 12m (with 95% binomial confidence interval).* |
| | Laboratory testing detects poliovirus in the sample: $P_{it}^{SampleTest}$ | Fixed | National | 0.9 (0.7 - 0.99) [37] |

## Relative risk of circulation per LGA

To accommodate differential risk of poliovirus circulation between LGAs, we adjust our calculation of sensitivity according to a relative risk based on data from the past 12 months. Our estimation of the underlying relative risk of circulation between LGAs followers the approach described by Voorman et al. [33,39], which incorporated estimated mucosal immunity, positive observations in the past 12 months (both AFP cases and environmental samples) and a radiation model of risk between neighbouring LGAs (suggested to reflect the dynamics of WPV spread in Nigeria better than distance or gravity models [40]). This relative risk is then transformed into an *adjusted* risk (AR) for each LGA, which is used to scale the estimated sensitivity in each LGA. In principle, inclusion of the adjusted risk means that increased surveillance effort in higher risk LGAs corresponds to a higher estimate of sensitivity than focussing this effort on lower risk LGAs.

## Overall system sensitivity

The sensitivity of each component across *all* LGAs (i.e., nationally) is one minus the probability of not detecting infection in *any* LGA (adjusted for variable underlying risk), given that infection is present at the design prevalence in at least one LGA. This is one minus the product of the probabilities that each LGA would not detect infection, $Pneg_{it}^{AFP/ENV}$:

$$Se_t^{AFP} = 1 - \prod_{i=1}^{I} \left( Pneg_{it}^{AFP} \right)$$

$$Se_t^{ENV} = 1 - \prod_{i=1}^{I} \left( Pneg_{it}^{ENV} \right)$$

where

$$Pneg_{it}^{AFP} = 1 - \left( Dp^{region} \cdot Se_{it}^{AFP} \cdot AR_{it} \right)$$

$$Pneg_{it}^{ENV} = 1 - \left( Dp^{region} \cdot Se_{it}^{ENV} \cdot AR_{it} \right)$$

If an LGA has no operating ES the $Se^{ENV}$ is equal to zero. The total, national-level sensitivity of the entire system is then

$$Se_t = 1 - (1 - Se_{it}^{AFP}) \cdot (1 - Se_{it}^{ENV})$$

This can be interpreted as the probability that the surveillance system would yield a positive observation if infection was present at the design prevalence in at least one LGA. This calculation makes a simplifying assumption of independence between the AFP and ENV systems. Due to the low symptomatic rate of polio infection, the overlap between infections detected via the environment and those having developed into AFP will be small, however correlation may be induced by external factors influencing both systems. This will be investigated via a simple correlation test of the estimated sensitivities.

To propagate uncertainty in the underlying step-wise probabilities, sensitivity is estimated across 1,000 draws from beta distributions fit to the central values and uncertainty intervals given in Table 1.

## Probability of freedom from infection

The probability of FFI at time $t$ ($P_t^{FFI}$) is defined as the probability that infection is below the design prevalence, given that no positive AFP cases or environmental samples have been observed in *any* LGA. This is conditional on the prior months with no positives, and the possibility of re-introduction.

Therefore, the probability of freedom from infection in month $t+1$, given infection status and surveillance sensitivity in month $t$ and incorporating risk of introduction, is defined as

$$P_{t+1}^{FFI} = (1 - P_t^{Infect})/(1 - P_t^{Infect} \cdot Se_t)$$

$P_t^{Infect}$ is the probability that the country was actually infected at time $t$, despite no positives being detected. This could be either because infections were missed and a false negative conclusion was drawn ($1 - P_t^{FFI}$), or because infection was re-introduced ($P^{Intro}$). As WPV1 at the time had been declared eliminated from all other African countries, and remained endemic only in Afghanistan and Pakistan, this risk of introduction was assumed to be low but non-zero. For this parameter we therefore defined a beta distribution centred on 1/1,000 with 95% quantile interval between 1/5,000 and 1/500.

The union probability of these two scenarios is

$$P_t^{Infect} = \left(1 - P_t^{FFI}\right) + P^{Intro} - \left(1 - P_t^{FFI}\right) \cdot P^{Intro}$$

$P_1^{FFI}$ is the assumed prior probability that infection is below the design prevalence, given that the last positive detection was observed in month 0. For the primary analysis we assume a beta distribution centred on 0.5 for this value, but explore alternatives in sensitivity analyses. For subsequent time steps, the prior probability is defined sequentially as the previously calculated probability $P_{t-1}^{FFI}$.

## Results

### Observed surveillance activity

**Detection of WPV1.** Wild poliovirus serotype 1 had been consistently detected among reported cases of AFP during the years prior to 2014 and occasionally detected in environmental samples. Between August 2014 and June 2016 (a period of 23 months), all observations from both AFP and ES were negative for WPV1 (Fig 2). In July 2016, two cases of AFP reported in the north-eastern state of Borno were found to be WPV1-positive. Genetic analysis showed little similarity between the isolates but identified a weak link to a virus last detected in the region in 2013, suggesting a prolonged period of undetected transmission rather than importation of infection [41]. Known security concerns, inaccessibility and population displacement in the region also supports this conclusion.

The 12-month rolling reported AFP rates among under-15s reflect extensive surveillance efforts during this time, with only a handful of LGAs not meeting the target rate of 2 per 100,000 for at least half of months (Fig 3A and 3B). In fact, the majority of LGAs were consistently exceeding the target rate by ten-fold or even higher. This is also the case for adequacy of stool sampling among reported AFP cases (Fig 3C and 3D).

A number of new environmental surveillance sites were established during this period, increasing the total number of LGAs with *any* active ES but decreasing the average population coverage per LGA (Fig 4). Although the median population size within 5km of a collected sample is 169,000 (Inter-Quartile Range; IQR [42,500–402,700]), within each LGA only 3% of the population is estimated to fall within the catchment of an active ES site in a given month on average. Exploring alternative catchment radii in sensitivity analyses, this percentage varies from 1.2% with a radius of 2km to 4.2% with a radius of 10km (*Fig E in* S1 Text).

Adequacy of sampling at ES sites in each LGA, with respect to detection of EV in collected samples, is more variable between LGAs than stool sampling adequacy among reported AFP cases (Fig 4C).

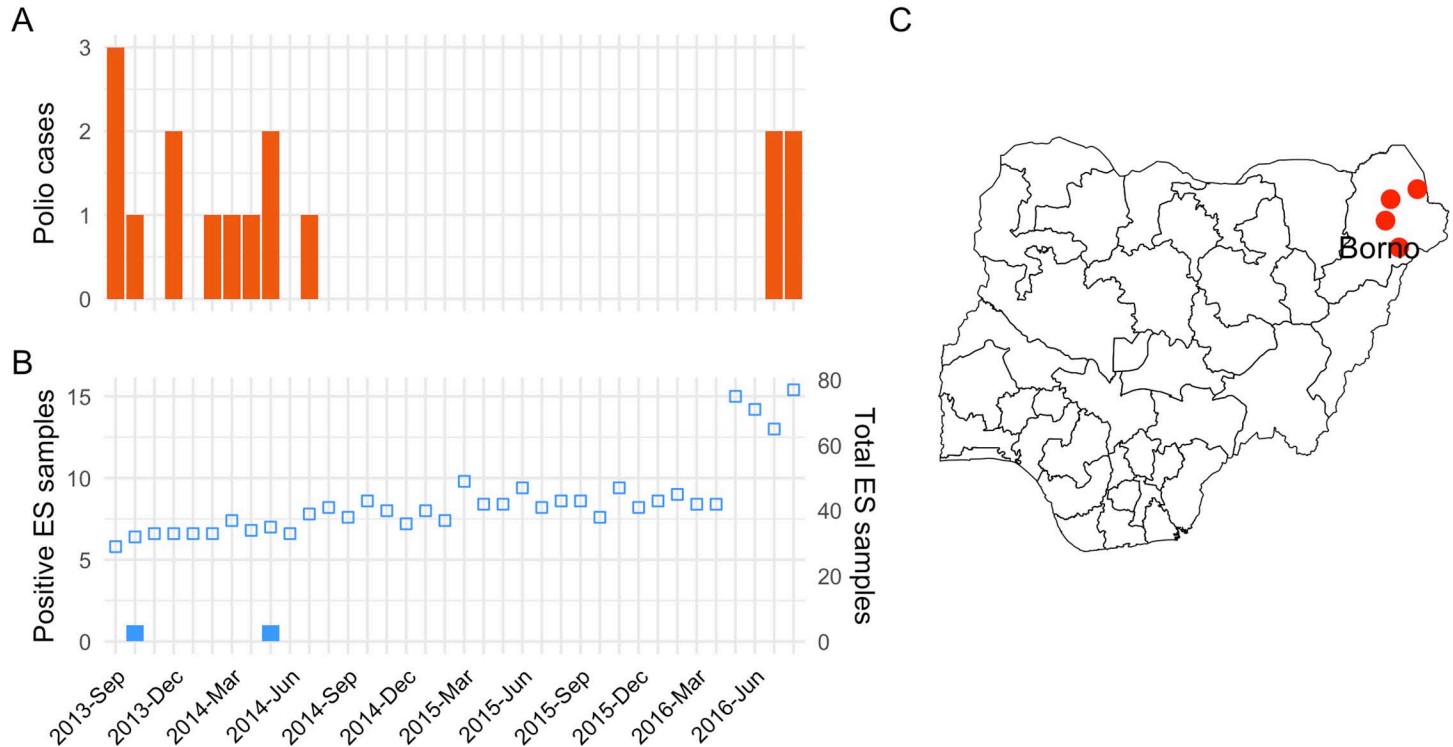

**Fig 2. Detections of wild poliovirus serotype 1 (WPV1) among cases of (A) AFP and (B) environmental surveillance (ES) samples.** The total number of environmental samples collected is indicated by unfilled blue squares. (C) The *recorded locations of four WPV1-positive cases of AFP detected in Borno state in July 2016, after 23 months of surveillance without detection (AFP cases are plotted randomly within LGA boundaries). Administrative boundaries are sourced from the Office for the Surveyor General of the Federation of Nigeria (OSGOF), Ehealth, United Nations Cartographic Section (UNCS) via the Humanitarian Data Exchange (*https://data.humdata.org/dataset/cod-ab-nga*), made available under the CC-BY-IGO license.*

## Estimating sensitivity and freedom from infection

The effect of increasing ES capacity over time and improved quality of AFP surveillance is observed in the resulting sensitivity estimates for the entire country (Fig 5A), which increases on average for both components between 2014 and 2016. However, these gains are minimal and the absolute probability of either component detecting infection (if prevalent at the design prevalence) remains small. This is to be expected given that AFP surveillance is limited by high asymptomatic infection rate, and ES by low overall coverage of the population. A difference in scale arises from the substantially greater uncertainty in the probability that an AFP case is notified, compared to any of the stepwise probabilities relating to ES. The greatest source of uncertainty in ES sensitivity likely arises from catchment; however, through sensitivity analysis we find that relatively large uncertainties in the catchment area translate to small differences in sensitivity (*Fig E in* S1 Text).

Considering the underlying LGA-specific estimates, the mean monthly sensitivity of ES is 2.6% per LGA (95% quantile interval: 0–39%) across all LGAs/months and 96% (65–100%) across only LGA-months with greater than 10% population coverage. A weak positive correlation is observed between AFP and ENV sensitivity per LGA and month during this period (Pearson's correlation and 95% confidence interval: 0.28 [0.27-0.30]).

By July 2016, when WPV1 was again detected in Borno state, we estimated a national probability of freedom from infection of 85.2% (95% uncertainty interval: 77.1-90.0%). Considering evidence from AFP surveillance alone this falls to 75% (64.7-83.0%) and ES alone to 65% (53.2-75.3%).

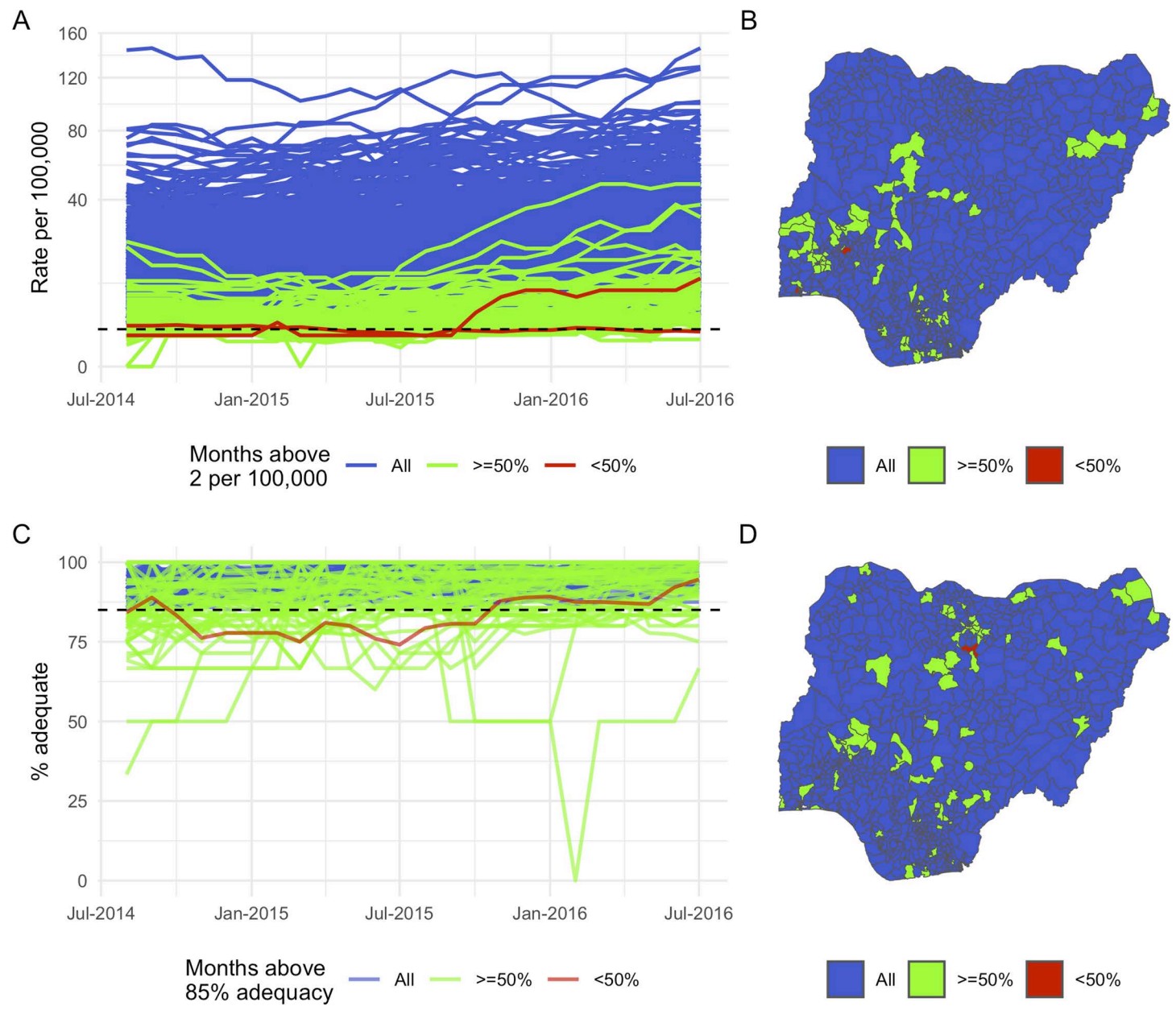

**Fig 3. Summary of observed AFP surveillance performance per LGA.** *(A,B) The 12-month rolling rate of reported AFP per LGA for the period of Aug 2013-July 2016. LGAs are classified by the proportion of time in which the reported rate exceeds the target threshold of 2 cases per 100,000 under-15s. (C,D) The 12-month rolling percentage of adequate stool sample collection among AFP cases reported by each LGA, in this case classified with respect to the target of 85%. In all panels, performance is classified into three levels according to how consistently the WHO target threshold is met; completely (blue), mostly (green) and partially (red). LGAs are mapped by this classification in the two right-hand figures. Administrative boundaries are sourced from the Office for the Surveyor General of the Federation of Nigeria (OSGOF), Ehealth, United Nations Cartographic Section (UNCS) via the Humanitarian Data Exchange (*https://data.humdata.org/dataset/cod-ab-nga*), made available under the CC-BY-IGO license.*

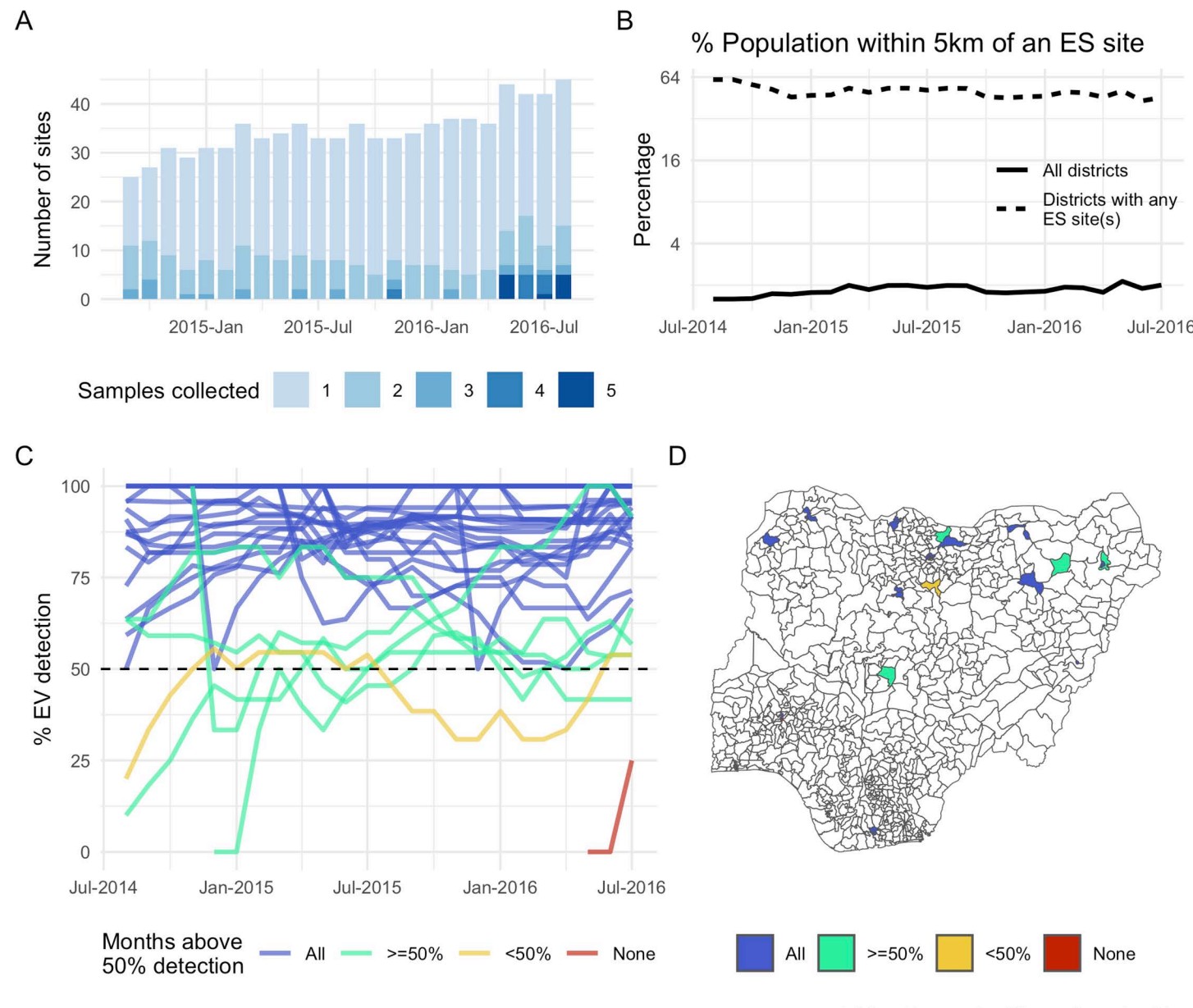

**Fig 4. Summary of observed ENV surveillance performance by LGA.** (A, B) *The coverage of ES by LGA and 12-month-rolling coverage over time.*
(C, D) The 12-month rolling detection rate of enterovirus (EV) among samples collected per LGA, classified with respect to the target of 50%. As in Fig
3, performance in panel C is classified according to how consistently the WHO target threshold is met, here into four levels; completely (blue), mostly
(green), partially (yellow) and not at all (red). LGAs are mapped by this classification in panel D. *Administrative boundaries are sourced from the Office
for the Surveyor General of the Federation of Nigeria (OSGOF), Ehealth, United Nations Cartographic Section (UNCS) via the Humanitarian Data
Exchange (*https://data.humdata.org/dataset/cod-ab-nga*), made available under the CC-BY-IGO license.*

## Declaration of elimination: 2016–2020

If we apply the same analysis to the subsequent period from 2016 to 2020, we find that the FFI probability that is attained
by the official declaration of WPV elimination in August 2020 [42,43] is 98.2% (97.5-98.5%). Our estimated trajectory for

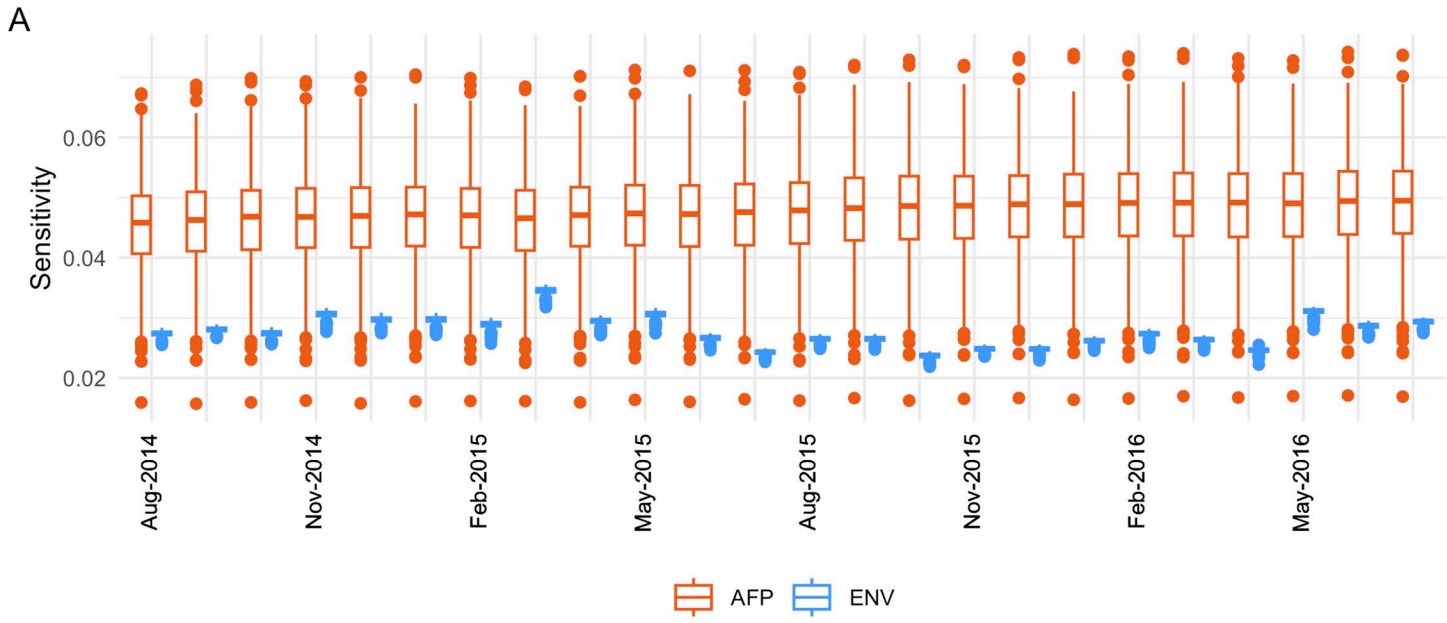

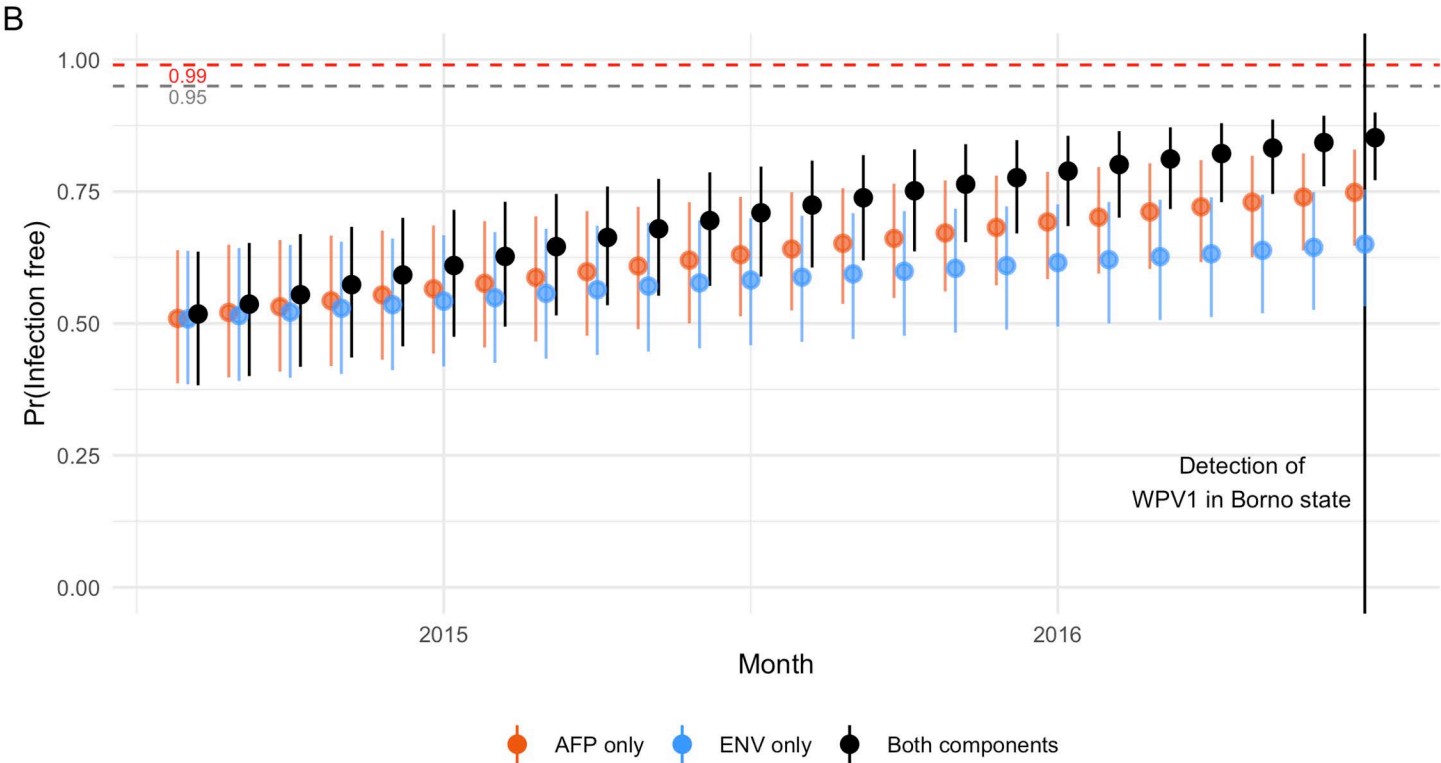

**Fig 5. (A) Estimated sensitivity of AFP (orange) and ENV (blue) surveillance per month, for detecting infection at the specified design prevalence of 1 per 100,000 in any LGA.** The box plots illustrate uncertainty across 1,000 draws for each probability in the scenario tree. (B) Inferred probability of freedom from infection for each accumulating month without detection of WPV1 through either AFP or ENV surveillance. Uncertainty is represented by 95% quantile intervals across 1,000 draws. Thresholds of 95% and 99% probability are illustrated with dashed lines. Results from the combined surveillance system are compared to those obtained by considering each component alone.

reaching the threshold of 95% is approximately two months shorter than Eichner and Dietz's estimated timeline of 2.9 years after the last observed case [1], which did not account for evidence from environmental surveillance (Fig 6B).

During this time we see a steady increase in the sensitivity of ENV surveillance (Fig 6A), as the network of sites was substantially expanded (*Fig C in* S1 Text). This is offset by a slight decrease in the sensitivity of AFP surveillance as the average adequacy of stool sample collection declined from 99% to 95% and the notification rate of AFP declined from 18 to 6.7 per 100,000 (*Fig D in* S1 Text). The sensitivity of ES overtakes AFP surveillance in mid-2018, and considering ENV surveillance alone results in a level of confidence that is approaching that of the combined system (91% (86.2-93.7%); Fig 6B) by August 2020.

## Sensitivity analyses

We considered two alternative assumptions about catchment radius around sampling sites - 2km and 10km. Of course, increasing the catchment size results in a larger estimated proportion of the population represented in ES and hence an increase in sensitivity. However, this difference was not large enough to have a substantial impact on the resulting elimination trajectory (*Fig E in* S1 Text).

Specifying a less stringent design prevalence of 1 per 10,000 resulted in a high probability of freedom from infection as early as mid-2015 (*Fig F in* S1 Text), which does not align with the known persistence of transmission during this time.

The assumed prior for the probability of freedom from infection immediately after the last observed positive does have a substantial influence on the inferred trajectory (*Fig G in* S1 Text). In this example, assuming either an ambivalent or more conservative (i.e., lower) distribution resulted in estimates more consistent with the timeline that was observed.

Not accounting for any variation in surveillance sensitivity over time results in an under-estimation of the evidence for freedom from infection in this case (*Fig H in* S1 Text), and a further five months until the estimated probability exceeds 95%.

## Discussion

Undetected transmission and asymptomatic infection are a fundamental concern for the global eradication of wild poliovirus. We may only define elimination with a lag from the last observed paralytic case - beyond the feasible limits of silent transmission - and the question of how long to wait is fundamental to defining the success of the programme and the safe withdrawal of the oral polio vaccine. Premature declaration of eradication based on insufficient evidence that transmission has ceased would have disastrous consequences, potentially setting back the Global Polio Eradication Initiative's strategic plan by decades.

In this analysis, we demonstrated a statistical framework within which routine indicators of surveillance performance feed into our interpretation of the absence of detected virus. We expand on previous applications of the approach with more spatially- and temporally-explicit scenario tree probabilities, informed by these indicators. We also explore more realistic assumptions than previous applications as to the population catchment of ES, based on the point location of active sites as opposed to a fixed district-level proportion. We estimate a monthly combined sensitivity of ES and AFP surveillance to detect infection across Nigeria that varies between 5–9% from 2014-16, and 6–12% from 2016-20. From this we are able to determine timelines for the declaration of regional elimination after the last reported case or positive environmental sample.

## Key findings

We evaluated our approach against known positive and negative scenarios of WPV1 elimination in Nigeria, and found that our estimated timelines for high confidence in freedom from infection were consistent with both the re-emergence of WPV1 in July 2016 and the official declaration of elimination four years later. This supports prospective use of the

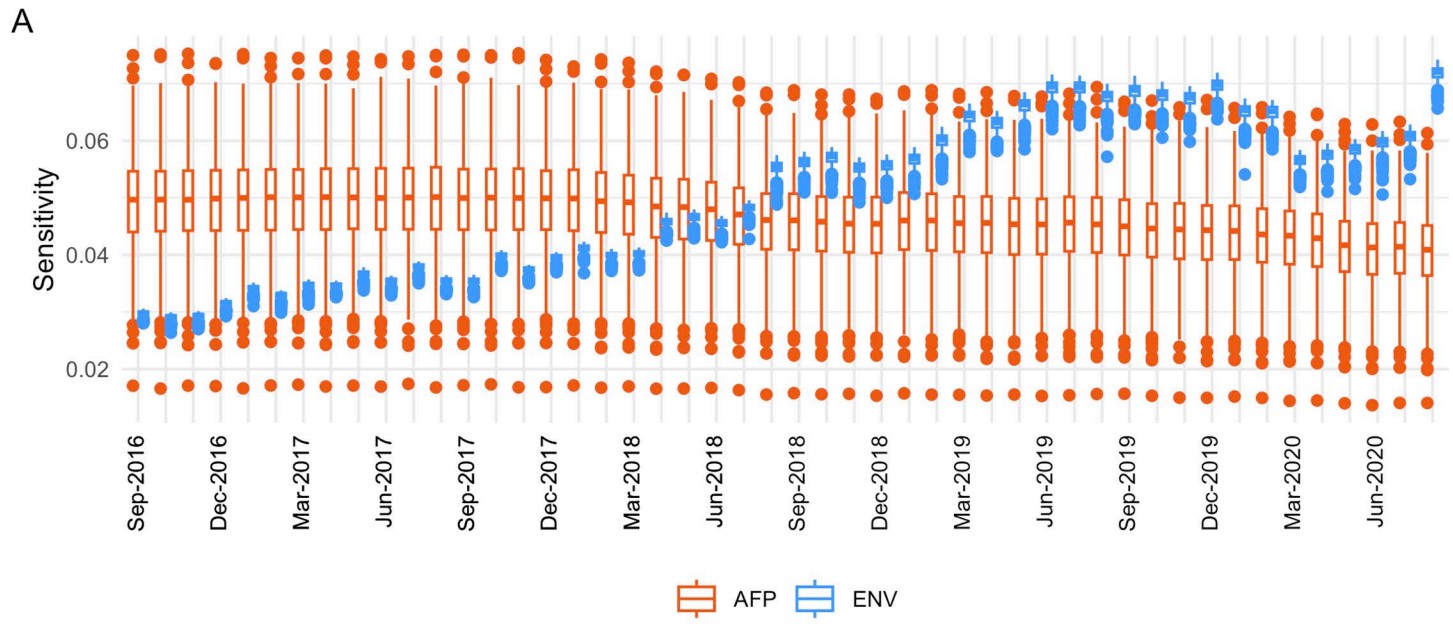

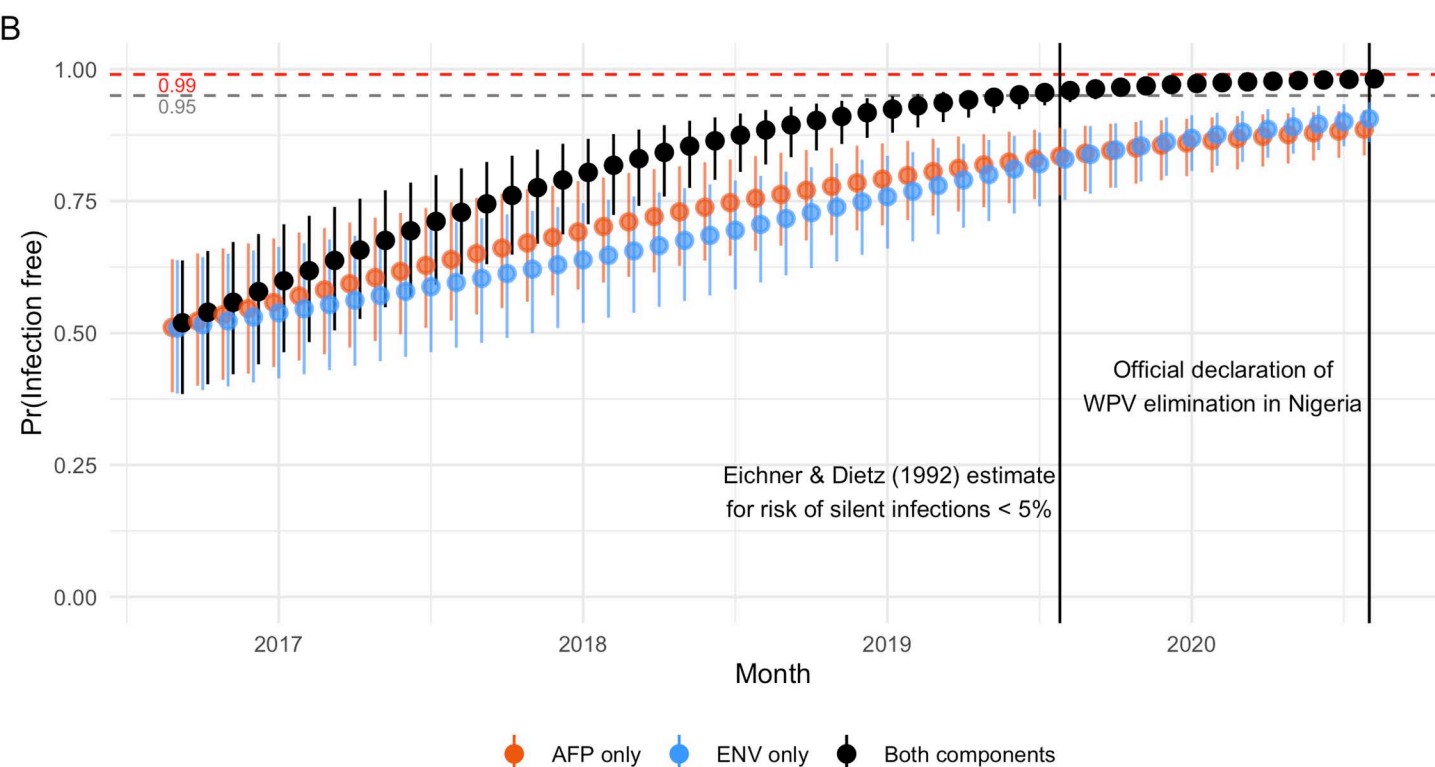

**Fig 6. Equivalent illustrations of estimated sensitivity and probability of freedom from infection as in Figure 5, for the subsequent period from 2016-2020.** The estimated timeline from Eichner and Dietz [1] for the risk of continued "silent" infections to fall below 5% after detection of the last clinical case is marked in (B), along with the official declaration of WPV elimination from Nigeria in August 2020.

approach to inform timelines for certification of the remaining WPV-endemic countries of Afghanistan and Pakistan, accounting for the specific characteristics of surveillance in those settings.

We further observe that, likely due to the expansion of ES after 2018, considering evidence from ES alone yields similar levels of confidence as pre-2018 data from the combined surveillance system. Our estimated trajectory for reaching the threshold of 95% is approximately two months shorter than Eichner and Dietz's estimated timeline (which did not incorporate evidence from environmental surveillance), a difference that may be increased with greater ES coverage. This emphasises the added value of ES relative to AFP surveillance in near-elimination settings. Especially given that the WHO now recommends the use of inactivated polio vaccine in the majority of settings [44] - which prevents *paralytic* poliomyelitis but not asymptomatic infection - extensive ES may be crucial to detect silent transmission.

Average monthly sensitivity of ES to detect infection is estimated at 3% between 2014–16, increasing to around 5% between 2016–20. These national level estimates (incorporating districts with no ES at all) fall considerably lower than some previous estimates. Kroiss et al. [6] looked at the rate of sabin virus detection in ES samples following SIAs on a district level, not accounting for the limited geographical catchment of sites within district boundaries. O'Reilly's approach in [4] estimated sensitivity on a population level - as we have done - but only within districts with active ES and those neighbouring. Our estimates are similar in magnitude to those obtained from applying the same scenario tree framework in Australia and in England and Wales [15,45], but there remain important differences in assumptions that limit comparability. Watkins et al. [15] only accounted for variation in AFP sensitivity between jurisdictions - not over time - and O'Reilly et al. [45] assumed a high ES population catchment of 80% given the large convergent sewerage systems in England and Wales.

Previous applications of this framework to poliovirus surveillance systems have used a design prevalence motivated by the global elimination target of 1 case of AFP per 100,000 under-15s [15,45], while an application to dengue fever chose a value corresponding to a single case among the entire population [24]. We found that specifying a less stringent design prevalence of 1 per 10,000 resulted in a high probability of freedom from infection as early as mid-2015, which does not align with the known persistence of transmission during this time. Overall, the consistency of our conclusions with what was observed in Nigeria between both 2014–16 and 2016–20 suggests that our assumptions for these two elements are reasonable.

## Limitations

Our estimation of AFP sensitivity is limited by uncertainty in the probability of a symptomatic case being reported. Nuance between LGAs is lost among consistently high AFP rates relative to the target of 2 per 100,000; it is not clear what factors drive the variability at higher levels, at what point higher rates do not translate to higher probability of reporting, or how high regional rates may mask care-seeking barriers in certain communities. In particular, the persistence of undetected WPV1 circulation in northeastern Nigeria was attributed to conflict and inaccessibility in the region at the time, which was expected to have disrupted surveillance, vaccination and health service utilisation [41,46], yet AFP rates remained substantially above target levels in the vast majority of LGAs. This suggests that important dynamics may be missed when interpreting the indicator against this minimal threshold. The extent to which the burden of conflict is evident across routine measures of surveillance performance could be investigated further, for example utilising publicly-available data on conflict events and fatalities from the ACLED project [47].

Inaccessibility is one of several external factors that could influence both AFP surveillance and ES, potentially inducing correlation that violates the simplifying assumption of independence and would lead to over-estimation of sensitivity. Some evidence of positive correlation was observed between our estimates for the two systems on an LGA level. Accommodating this dependence could therefore slightly lengthen our estimated timelines for elimination, but not to an extent that would contradict our conclusions. The viability of this assumption is, however, something that should be reconsidered for any new setting in which the approach is applied.

Our conclusions are sensitive to some other key assumptions. First, the model accounts for dependence between under-15s within the same LGA, but not for clustering at a lower level (e.g., within schools). As with correlation between the two surveillance systems, this could result in over-estimation of sensitivity. Second, the time to accumulate high probability of freedom from infection is strongly influenced by both the assumed prior probability and by the critical prevalence level we require the surveillance system to detect (the design prevalence). Here we chose a prior distribution for the FFI probability that was ambivalent (symmetrical and centred on 50%), yet this could be considered optimistic if a positive was observed in the previous month. We would argue that this depends on whether the last detection was - for example - the tail of a recent epidemic or the final blip in a stuttering chain, and suggest that the prior is specified in context of that recent pattern. Experts on the specific epidemiological context of a country or district setting may have a different view on what would be an appropriate prior. Further work to elicit and incorporate expert opinion (for example, via a Delphi study [48]) would add validity to the choice of assumption.

**Implications for future research**

For environmental surveillance, the greatest limiting factor for sensitivity is population catchment. A previous application of this framework [4] assumed a static catchment of around 60% of the population for each sub-region with active ES, and guidelines suggest that sites should be placed to have a catchment of around 100,000 people. This target is met by most collection sites when assuming a catchment radius of 5km. However, if we consider LGAs with either inconsistent sampling (i.e., some zero months) or no ES at all, only around 3% of an LGA's population is captured on average per month across the country. This equates to approximately 12,500 people, increasing to 19,000 people between 2016–20. The assumed 5km radius is also arguably generous in comparison to another analysis of ES performance in Nigeria [5]. Sensitivity analyses considering alternative catchments indicate that the area would have to be expanded to an unrealistic extent to yield any consequential increase in sensitivity. More realistic catchment areas may be inferred from geographic contours and water flow directions [5,49], but this work is yet to be adequately linked to the sample locations recorded in POLIS.

Having ES within sewage treatment works would likely increase the size of catchment areas. However, households connected to formal sewage networks are thought to be at lower risk of being infected with poliovirus (due to the prominence of faecal-oral transmission in the virus' persistence), and likely to have a higher income along with other socioeconomic advantages [50,51]. This bias would be expected in most polio-affected settings in which this framework could be applied [52] and, moreover, many of these are low-income settings with limited sewage infrastructure to begin with. Especially in informal sewage systems, the water flow, dilution, temperature, chemical contamination and recent weather will also influence what is detected in a collected sample [53,54]. The detection rate of EVs likely reflects some of this natural variability and serves as a proxy in our analysis, but routine measurement of key indicators during collection would allow future analyses to account for these sources of variability directly.

We suggest that this approach may also be applied to interpret the resolution of vaccine-derived poliovirus (VDPV) outbreaks, in particular those of type 2 (cVDPV2) which became widespread following the removal of serotype 2 from standard oral vaccination [55]. Lower symptomatic rates for serotype 2 would mean that AFP surveillance would be substantially less sensitive for cVDPV2 relative to ENV surveillance. This means that greater effort invested in consistent and well-distributed environmental sampling would be necessary to declare with confidence that a cVDPV2 outbreak is over. Such an analysis was conducted and presented to the GCC in November 2024 [56], highlighting the sparse coverage of environmental surveillance in several countries currently experiencing cVDPV2 outbreaks, relative to the coverage in Nigeria for the near-WPV1-elimination period presented here.

**Conclusion**

The GGC have agreed that the definition of timelines for declaring elimination of poliovirus should be data-driven and informed by the performance of surveillance in the specific setting of interest. This performance should ideally be

evaluated quantitatively in terms of statistical sensitivity to detect infection on a population level since - as we observe in this example - the meeting of predefined threshold levels may be insufficient to highlight blind spots in surveillance. Using the approach demonstrated here, temporal changes and localised disruption to surveillance may directly and quantitatively inform our interpretation of the absence of detection, and the accumulating evidence that transmission has been interrupted. This inference may be extended to data from Pakistan and Afghanistan to serve WPV eradication goals, as well as regions of sub-Saharan Africa and Southeast Asia to determine the resolution of cVDPV2 outbreaks. However, there is a need to more thoroughly define the impact of variable conditions of environmental sample collection and the disruption of AFP surveillance caused by conflict, in terms of the probability of persisting virus circulation being detected. With the conclusion of 40 years of global commitment in sight but balancing on the edge of significant resurgence, this level of detail in inference will be crucial to confidently certify the last endemic areas as free from wild poliovirus.

## Supporting information

**S1 Text. Including: Section A.** Scenario tree probability definitions: Description of evidence and assumptions behind the scenario tree model of surveillance sensitivity. **Fig A:** Frequency of sample collection per ES site per month, from Sep 2014 - Sep 2016. From May 2016 frequency increased at many sites, alongside an increase in the number of active sites overall. **Fig B: (A)** Example simulations of shedding durations for a single infected individual within an ES catchment in a given month (30 days), and the coincidence of a random monthly sample collection. **(B)** Simulated probabilities of sample collection coinciding with the period of shedding of a single infected individual within the catchment, for assumed sampling frequency from 1-30 per 30 days. These distributions are then used to determine the scenario tree probability of sample collection during shedding, given the recent frequency of sampling observed at sites in a certain LGA. Section B. Surveillance activity 2016–2020. **Fig C:** Equivalent to Fig 3 in the main text, but for the period 2016–2020. **Fig D:** Equivalent to Fig 4 in the main text, but for the period 2016–2020. Section C. Sensitivity analyses: **Fig E(A) (B):** Resulting probabilities of freedom from infection given different assumptions for the catchment radius around ES sites. Expanding the radius up to 10km (highly unrealistic) yields marginal gains compared to the primary assumption of 5km. **Fig F: (A)** Estimated sensitivity of the two surveillance components across all LGAs, varying the assumed design prevalence (the minimum WPV1-infection prevalence to be detected) from 1 per 1,000–1 per 100,000. In particular for AFP surveillance, sensitivity is substantially higher if we only require the system to detect as low as one infection per 1,000 population. The difference is smaller for ENV surveillance, as it is able to detect infection directly as opposed to only clinical disease. **(B)** Resulting probabilities of freedom from infection given different assumptions for the design prevalence. Shaded areas highlight where the probability of freedom from infection has exceeded 0.95. Assuming a higher (i.e., less stringent) design prevalence results in a much quicker accumulation of evidence that infection is below this specified level, reaching the elimination threshold within only four months for 1 per 1,000 and around 12 months for 1 per 10,000. This is not consistent with what we know in retrospect, that transmission persisted undetected during this time. **Fig G:** Estimates trajectories of the probability of freedom from infection, according to two alternative assumed distributions for the prior probability at the first time point without detection (assumed distribution for the primary analysis - a Beta distribution with $\alpha = 30$ and $\beta = 30$ - shown in blue). Regardless of which prior was assumed, the estimated FFI probability had exceeded 95% before the official declaration of elimination in August 2020. **Fig H:** Comparison of inferred probability of freedom from infection, with and without taking into account varying sensitivity of surveillance over time. For the period 2016–2020, environmental surveillance was expanded and as a result the overall sensitivity of surveillance increased. Not accounting for this increase underestimates the accumulating evidence of freedom from infection, and results in an additional five months until the FFI probability exceeds 95%.
(DOCX)

## Acknowledgments

The authors would like to thank Arie Voorman and Hil Lyons for their role in the previous analysis on which this research is built.

## Author contributions

**Conceptualization:** Emily S. Nightingale, William John Edmunds, Kathleen M. O'Reilly.

**Data curation:** Emily S. Nightingale, Ly Pham-Minh, Isah Mohammad Bello, Tesfaye Bedada Erbeto, Megan Auzenbergs.

**Formal analysis:** Emily S. Nightingale.

**Funding acquisition:** Kathleen M. O'Reilly.

**Investigation:** Ly Pham-Minh, Samuel Okiror, Marycelin Baba, Adekunle Adeniji.

**Methodology:** Emily S. Nightingale, Kathleen M. O'Reilly.

**Software:** Emily S. Nightingale, Kathleen M. O'Reilly.

**Supervision:** William John Edmunds, Kathleen M. O'Reilly.

**Validation:** Emily S. Nightingale.

**Visualization:** Emily S. Nightingale.

**Writing – original draft:** Emily S. Nightingale.

**Writing – review & editing:** Ly Pham-Minh, Isah Mohammad Bello, Samuel Okiror, Tesfaye Bedada Erbeto, Marycelin Baba, Adekunle Adeniji, Megan Auzenbergs, William John Edmunds, Kathleen M. O'Reilly.

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
