## [Decision Letter · Decision Letter 0]

3 Jul 2025

Sub-national estimation of surveillance sensitivity to inform declaration of disease elimination: A retrospective validation against the elimination of wild poliovirus in Nigeria

PLOS Computational Biology

Dear Dr. Nightingale,

Thank you for submitting your manuscript to PLOS Computational Biology. After careful consideration, we feel that it has merit but does not fully meet PLOS Computational Biology's publication criteria as it currently stands. Therefore, we invite you to submit a revised version of the manuscript that addresses the points raised during the review process.

Please submit your revised manuscript within 60 days Sep 02 2025 11:59PM. If you will need more time than this to complete your revisions, please reply to this message or contact the journal office at ploscompbiol@plos.org. Please include the following items when submitting your revised manuscript:

We look forward to receiving your revised manuscript.

Kind regards,

Juliette Paireau

Academic Editor

PLOS Computational Biology

Benjamin Althouse

Section Editor

PLOS Computational Biology

**Journal Requirements:**

Potential Copyright Issues:

i) Figures 2B, 3B, 3D, 4D, S3B, S3D, and S4D. Please (a) provide a direct link to the base layer of the map (i.e., the country or region border shape) and ensure this is also included in the figure legend; and (b) provide a link to the terms of use / license information for the base layer image or shapefile. We cannot publish proprietary or copyrighted maps (e.g. Google Maps, Mapquest) and the terms of use for your map base layer must be compatible with our CC BY 4.0 license.

1) State the initials, alongside each funding source, of each author to receive each grant. For example: "This work was supported by the National Institutes of Health (####### to AM; ###### to CJ) and the National Science Foundation (###### to AM).".

**Reviewers' comments:**

Reviewer's Responses to Questions

**Comments to the Authors:**

**Please note that two reviews are uploaded as attachments.**

Reviewer #1: Reviewer comments for PCOMPBIOL-D-25-00277

This is a well-written manuscript which seeks to quantify the sensitivity of Nigeria’s polio surveillance system and derive a statistical framework which provides a probability of freedom from infection (FFI). This is then retrospectively validated against two periods of empirical absence of WPV1. The principal findings from the framework are concordant with the true virus transmission status and aim to provide a quantitative, spatially specific decision aid in polio eradication scenarios.

I think this manuscript is suitable for publication in PLoS Comp. Biol. after a few minor revisions.

Abstract

1) The opening sentence is a bit clumsy and could probably do with being split into two, something like:

‘Which duration of absence of polio infection qualifies as regional elimination is a key question in the global drive towards eradication of the disease. The safe withdrawal of the polio vaccine is naturally contingent upon this.’

2) Perhaps expand on ‘sensitivity over time and space’, introduce the time and spatial scales (as you have in the methods) more explicitly here so the reader can get a better feel for the size of the study.

Introduction

1) The independence between cases of AFP and detection of the virus via ES has been assumed, is there a suitable reference for this? A couple of lines stating/explaining this, as you did with vaccination would be helpful.

Methods

1) It would be good to include a summary of the sensitivity analyses you’ve conducted for some of the key assumptions (such as the design prevalence, prior FFI and ES catchment) in the main text rather than just stating you’ve done them. Please also refer to the specific item in the supplement (which for Plos comp biol is something like ‘Fig B in S1 Text’).

2) Treating AFP and ES as independent will over-estimate combined sensitivity should these two variables be influenced by the same external factors, might be worth mentioning or a simple correlation test.

3) Citation for lab test sensitivity?

4) Minor typo in ‘LGAs followers the approach’

Results

Figures look great and their message is clear. My only comment here is that it’s quite an ‘explosion of colour’ in some of the panels, perhaps describing the lines/colours in the legend could be helpful, particularly for colourblind readers.

Discussion

1) Could contrast your findings more explicitly with Dietz and Eichner’s 3-year rule.

2) I think the suggestion of ES superseding AFP surveillance might need further caveating, they’re fundamentally different things and the ability of ES to supersede may not be relevant in all foci.

3) The overall structure of the discussion gets a little jumpy and some of the paragraphs are long. Consider breaking it up into a more organised flow i.e. Implications – Limitations – Future work

Conclusion

1) GCC not GGC in the opening line

2) A concise statement about broader generalisability outside of Nigeria (Pakistan, Afghanistan etc) could provide a nice way to strengthen the conclusion.

Reviewer #2: Review attached.

Reviewer #3: There is a review uploaded as an attachment 'comments'

**Have the authors made all data and (if applicable) computational code underlying the findings in their manuscript fully available?**

Reviewer #1: Yes

Reviewer #2: Yes

Reviewer #3: Yes

PLOS authors have the option to publish the peer review history of their article (what does this mean? ). If published, this will include your full peer review and any attached files.

**Do you want your identity to be public for this peer review?** For information about this choice, including consent withdrawal, please see our Privacy Policy .

Reviewer #1: No

Reviewer #2: No

Reviewer #3: No

**Figure resubmission:**

**Reproducibility:**



---

## [Decision Letter · Decision Letter 1]

11 Dec 2025

PCOMPBIOL-D-25-00277R1

Sub-national modelling of surveillance sensitivity to inform declaration of disease elimination: A retrospective validation against the elimination of wild poliovirus in Nigeria

PLOS Computational Biology

Dear Dr. Nightingale,

Thank you for submitting your manuscript to PLOS Computational Biology. After careful consideration, we feel that it has merit but does not fully meet PLOS Computational Biology's publication criteria as it currently stands. Therefore, we invite you to submit a revised version of the manuscript that addresses the points raised during the review process.

We look forward to receiving your revised manuscript.

Kind regards,

Benjamin Althouse

Section Editor

PLOS Computational Biology

Benjamin Althouse

Section Editor

PLOS Computational Biology

**Additional Editor Comments:**

The paper is ready to be accepted. Please revise the typos pointed out by the reviewer and it will be ready.

**Journal Requirements:**

1) We have noticed that you have uploaded Supporting Information files, but you have not included a complete list of legends. Please add a full list of legends for your Supporting Information files after the references list.

2)  We strongly recommend all authors deposit their data before acceptance, as the process can be lengthy and hold up publication timelines. Please note that, though access restrictions are acceptable now, your entire minimal dataset will need to be made freely accessible if your manuscript is accepted for publication. This policy applies to all data except where public deposition would breach compliance with the protocol approved by your research ethics board. If you are unable to adhere to our open data policy, please kindly revise your statement to explain your reasoning and we will seek the editor's input on an exemption.

**Reviewers' comments:**

Reviewer's Responses to Questions

**Comments to the Authors:**

Reviewer #2: Uploaded as an attachment.

**Have the authors made all data and (if applicable) computational code underlying the findings in their manuscript fully available?**

Reviewer #2: Yes

PLOS authors have the option to publish the peer review history of their article (what does this mean? ). If published, this will include your full peer review and any attached files.

**Do you want your identity to be public for this peer review?** For information about this choice, including consent withdrawal, please see our Privacy Policy .

Reviewer #2: No

**Figure resubmission:**
---

## [Editor Report · Decision Letter 2]

4 Feb 2026

Dear Dr Nightingale,

We are pleased to inform you that your manuscript 'Sub-national modelling of surveillance sensitivity to inform declaration of disease elimination: A retrospective validation against the elimination of wild poliovirus in Nigeria' has been provisionally accepted for publication in PLOS Computational Biology.

Best regards,

Benjamin Althouse

Section Editor

PLOS Computational Biology

---

## [Editor Report · Acceptance letter]

PCOMPBIOL-D-25-00277R2

Sub-national modelling of surveillance sensitivity to inform declaration of disease elimination: A retrospective validation against the elimination of wild poliovirus in Nigeria

Dear Dr Nightingale,

I am pleased to inform you that your manuscript has been formally accepted for publication in PLOS Computational Biology. Your manuscript is now with our production department and you will be notified of the publication date in due course.

With kind regards,

Aiswarya Satheesan
